# Patient-Reported Outcomes in Cancer Patients with HIV

**DOI:** 10.3390/cancers14235889

**Published:** 2022-11-29

**Authors:** Anna E. Coghill, Naomi C. Brownstein, Sweta Sinha, Zachary J. Thompson, Brittney L. Dickey, Aasha I. Hoogland, Peter A. Johnstone, Gita Suneja, Heather S. Jim

**Affiliations:** 1Center for Immunization and Infection Research in Cancer, H. Lee Moffitt Cancer Center & Research Institute, Tampa, FL 33612, USA; 2Cancer Epidemiology Program, H. Lee Moffitt Cancer Center & Research Institute, Tampa, FL 33612, USA; 3Biostatistics and Bioinformatics Program, H. Lee Moffitt Cancer Center & Research Institute, Tampa, FL 33612, USA; 4Health Outcomes & Behavior Program, H. Lee Moffitt Cancer Center & Research Institute, Tampa, FL 33612, USA; 5Department of Radiation Oncology, H. Lee Moffitt Cancer Center & Research Institute, Tampa, FL 33612, USA; 6Department of Radiation Oncology, University of Utah, Salt Lake City, UT 84108, USA

**Keywords:** HIV, HIV and cancer survivorship, patient-reported outcomes

## Abstract

**Simple Summary:**

Cancer is becoming more common as people living with HIV (PWH) survive to older ages due to the use of effective HIV treatments. Due to this, cancer is now a leading cause of death in PWH. In this study, we used three years of data from Moffitt Cancer Center to try and understand the cancer experience in PWH, including whether cancer patients with HIV report different symptoms during cancer care compared to the general cancer population. In order to accomplish this, self-reported symptom data were collected from 12,529 patients, including 55 with HIV. We found that cancer patients with HIV were nearly 50% more likely to experience poor symptoms during their cancer care, including severe pain and depression. In addition, the presence of these symptoms was linked to the likelihood of patients dying. These findings should prompt future projects to help manage poor symptoms during cancer care for PWH.

**Abstract:**

Elevated cancer-specific mortality in PWH has been demonstrated for non-AIDS-defining malignancies. However, additional clinical endpoints of interest, including patient-reported outcomes (PROs), have not been systematically examined in PWH and cancer. We evaluated differences in patient-reported symptomology between cancer patients with versus without HIV using data from 12,529 patients at the Moffitt Cancer Center, including 55 with HIV. The symptoms were assessed using the Edmonton Symptom Assessment Scale (ESAS), which asks patients to rank 12 symptoms on a scale of 1–10, with scores ≥7 considered severe. The responses across all questions were summed to create a composite score. Vital status through t July 2021 was determined through linkage to the electronic health record. PWH reported a higher composite ESAS score on average (44.4) compared to HIV-uninfected cancer patients (30.7, *p*-value < 0.01). In zero-inflated negative binomial regression models adjusted for cancer site, sex, and race, the composite ESAS scores and the count of severe symptoms were 1.41 times (95% CI: 1.13–1.77) and 1.45 times (95% CI: 1.09–1.93) higher, respectively, in cancer patients with HIV. Among PWH, higher ESAS scores were associated with mortality (*p*-value = 0.02). This is the first demonstration of uniquely poor PROs in PWH and cancer and suggests that patient symptom monitoring to improve clinical endpoints deserves further study.

## 1. Background

The widespread use of effective antiretroviral therapy has translated into increased longevity for persons living with HIV (PWH). This has changed their co-morbidity profile, a fact reflected by the growing burden of chronic diseases, including cancer [1]. Cancer is now the leading cause of non-AIDS death in PWH, and mortality following a cancer diagnosis is elevated in PWH compared to the general oncology population [2,3,4]. Potential contributors to this unequal cancer mortality burden include later stage at diagnosis [5] and barriers to cancer treatment initiation [6,7], although population and hospital-based national registry data demonstrate that HIV-associated cancer survival disparities persist after adjustment for stage and treatment initiation among older PWH [3]. This association between patient HIV status and increased mortality after a cancer diagnosis prompted us to evaluate a broader spectrum of clinical endpoints. 

Patient-reported outcomes (PROs) are defined by the US Food and Drug Administration as “any report of the status of a patient’s health condition that comes directly from the patient, without interpretation of the patient’s response by a clinician or anyone else” [8,9]. PROs have been linked to a number of objective clinical outcomes, including response to therapy, cancer progression, and survival in the general (i.e., HIV-uninfected) oncology population [10,11,12,13,14,15]. Although PROs are increasingly recognized as important endpoints to capture during cancer therapy, we are not aware of any systematic descriptions of PROs in cancer patients with HIV using real-world data. A precedent exists to suggest that PWH may experience cancer therapy differently. Reports from the US Veterans Administration (VA) indicate that anal cancer patients with HIV are more likely to present with severe, grade 3–4 hematologic adverse events and more frequent hospitalizations during chemoradiation [16,17]. PROs provide information that is complementary to the clinicians’ ratings of adverse events, and we posit that PROs during cancer care differ by HIV status. 

## 2. Methods 

We ascertained patient-reported symptomology in individuals who received care at Moffitt Cancer Center, an NCI-designated comprehensive cancer center in southwest Florida, between February 2017 and July 2020. Self-reported symptom data were collected using the Edmonton Symptom Assessment Scale (ESAS) [18]. The ESAS is comprised of 12 questions that ask patients to rank each symptom on a scale ranging from 0 to 10, with responses ≥7 classified as severe (Supplement 1). Symptoms include anxiety, constipation, depression, difficulty sleeping, drowsiness, lack of appetite, overall well-being, nausea, pain, shortness of breath, spiritual well-being, and tiredness. The ESAS is administered to patients in the Radiation Oncology, Head and Neck Oncology, and Supportive Care Medicine clinics at Moffitt. For each participant, we computed (1) a composite score that summed responses across all 12 symptoms and (2) a severity score that counted responses ≥7. HIV status was determined by presence of International Classification of Disease (ICD) codes indicative of HIV (ICD-9: 042-044, 079.53, 795.71, 795.8; ICD-10: V08, B20, B97.35, R75, O98.7, Z21) in the Moffitt health research information database. HIV-associated differences in each ESAS score were tested using a Wilcoxon rank sum test. 

For adjusted analyses, we produced zero-inflated negative binomial (ZINB) regression models examining the association between ESAS scores and HIV, adjusted for cancer site (anal, hematopoietic, other solid organ tumors), biological sex as noted in the patient medical record, and race (Black, White, other) as reported to the cancer registry, as these three variables were each associated with ESAS scores in univariate analyses. Using the Vuong test, we confirmed that the ZINB model best fit the data, evidence that the cancer patient population in this study consisted of two subpopulations, one likely to report no symptoms (zero inflation component), and one likely to report a range of symptoms (negative binomial component). 

Linkage to the electronic health record (EHR) for PWH provided vital status information through July 2021. The association of ESAS scores, categorized into tertiles based on the distribution within our study population, with all-cause mortality was examined using a chi-square log-rank test for trend. We also performed a sensitivity analysis of the association between mortality and ESAS scores restricted to patients with early stage disease. For this analysis that was limited by sample size, ESAS scores were parameterized as above/below our study population median. All analyses were produced in R version 3.6.3 or PRISM version 8. De-identified data are available from the corresponding author (A.E.C.) upon reasonable request and documentation of necessary ethical research approvals.

## 3. Results 

This report includes data from 12,529 Moffitt Cancer Center patients, 55 of whom had received an HIV diagnosis. PWH were younger (mean age = 52 years) than the HIV-uninfected individuals (mean age = 62 years). PWH were also more likely to be male (PWH: 80.0%; HIV-uninfected: 54.1%) and non-White (PWH: 41.8%; HIV-uninfected: 14.3%). The majority of patients, regardless of HIV status, had undergone a documented receipt of cancer treatment (PWH: 87.3%; HIV-uninfected: 90.6%). The most common diagnoses among PWH were anal cancer and hematopoietic malignancies (27.3% and 14.5% of cancers in PWH, respectively). In contrast, nearly all (>90%) of cancer patients without HIV were diagnosed with solid organ tumors, with the most common being cancers of the breast (17%), prostate (17%), head and neck (16%), and lung (14%). Despite this range of tumor types, the proportion of patients with receipt of radiation (PWH: 16%; HIV-uninfected: 19%) or chemotherapy (PWH: 11%; HIV-uninfected: 9%) as part of their ongoing cancer care did not differ by HIV status. PWH were less likely to receive surgery as part of current cancer care (PWH: 13%; HIV-uninfected: 25%). Regarding exposure assessment, PWH answered an average of 4.9 ESAS questionnaires (range = 1–28), and HIV-uninfected patients similarly completed an average of 4.2 (range = 1–48) questionnaires. The worst recorded ESAS score for each symptom was used for this report. Slightly more than one-third of all patients (PWH: 35%; HIV-uninfected: 38%) reported their worst symptom score on a day during which they were actively undergoing cancer treatment. 

Overall, PWH reported a higher mean composite symptom score compared to HIV-uninfected cancer patients (PWH: 44.4; HIV-uninfected: 30.7, *p*-value < 0.01, Figure 1A)**.** This pattern was observed for all 12 symptoms, with the most pronounced HIV-associated symptom differences (≥1.5) observed for pain, overall well-being, and difficulty sleeping (Figure 1B). Symptom differences ≥1.0 were reported for depression, constipation, spiritual well-being, and tiredness, while differences ranged from 0.6 to 0.9 for lack of appetite, nausea, shortness of breath, drowsiness, and anxiety. No symptoms ascertained using the ESAS tool were more common in HIV-uninfected cancer patients (Supplement 2). 

We next performed a sensitivity analysis restricted to patients who reported their worst ESAS symptom score on a day they were actively undergoing cancer treatment (*n* = 4744). The composite symptom score remained statistically significantly higher in PWH (PWH: 41.4; HIV-uninfected: 30.5, *p*-value = 0.04). The ESAS summary scores were also statistically significantly higher when we restricted the analysis to the 276 patients diagnosed with anal cancer, the most common tumor among PWH at our institution during this timeframe (PWH: 52.1; HIV-uninfected: 34.2, *p*-value = 0.01).

The results from adjusted regression analyses confirmed that PWH reported a higher frequency and severity of poor symptoms during cancer care compared to HIV-uninfected cancer patients treated at our institution, even after accounting for HIV-related differences in cancer site, biological sex, and race recorded in the cancer registry. In patients that reported at least one symptom (*n* = 11,668; 93.1%), the composite symptom score and the number of severe symptoms were 1.41 times (95% CI: 1.13–1.77) and 1.45 times (95% CI: 1.09–1.93) higher, respectively, in PWH compared to cancer patients without HIV (Table 1). The likelihood of reporting no symptoms was similar by HIV status (OR: 1.09 [95%CI: 0.37–3.24]). However, this outcome was observed to be different for anal cancer patients, who were over-represented among PWH. Specifically, anal cancer patients were less likely to report zero symptoms (OR: 0.39 [95% CI: 0.17–0.91]) and less likely to report no severe symptoms (OR: 0.50 [95% CI: 0.26–0.97]) compared to patients diagnosed with either hematopoietic malignancies or other solid organ tumors.

Among the 55 PWH, 22 deaths were recorded. Those with a higher composite symptom score and those with a higher number of severe symptoms were more likely to be recorded as deceased in the EHR during the study period (log-rank *p*-values = 0.02 and *p* = 0.03, respectively). In a sensitivity analysis restricted to the 19 PWH with stage I-II disease, who may be expected to have the most favorable symptom profile, we further examined the association between mortality and ESAS scores. Despite limited sample size, we observed evidence that PWH with higher composite and severe scores remained more likely to be recorded as deceased in the EHR during the study period (log-rank *p*-values = 0.10 and *p* = 0.02, respectively).

## 4. Discussion

Among cancer patients who reported symptoms during cancer care, those with HIV reported a higher frequency and severity of symptoms compared to cancer patients without HIV from the same institution. This PROs difference persisted after adjustment for cancer site, sex, and race, and was consistently observed across the 12 distinct symptoms queried. Additionally, we observed in this study that self-reported symptomology among PWH was associated with an increased risk of all-cause mortality.

To our knowledge, this is the first study systematically comparing PROs between cancer patients with versus without HIV using real-world data. The increasing cancer burden in PWH is well documented [1], as is the association between HIV and cancer-specific mortality [2,3]—cancer is now the leading cause of death in the aging HIV population. Research from the VA has observed higher rates of clinician-assessed hematologic toxicity and increased hospitalizations during chemoradiation among anal cancer patients with HIV [7,16]. Our data expand on these prior reports and suggest a compelling difference in the patient treatment experience in PWH and cancer that includes a higher symptom burden during cancer care.

The PRO differences we report are potentially clinically important. PROs are now frequently incorporated as key data points in randomized clinical trials [19,20,21] in oncology and have recently been cited in a meta-analysis that suggests certain therapeutic choices as being more tolerable due to a limited PROs profile [22]. Notably, the patient treatment experience can be actively monitored using validated PROs assessment tools, including readily available electronic monitoring between clinic visits [23]. In light of the data from the general cancer population demonstrating that clinic-based patient symptom monitoring can significantly improve quality of life and cancer patient survival [24,25], our findings should prompt formal investigations of patient symptom management strategies specific to PWH and cancer to address their documented cancer survival disparity.

Strengths of this study include its efficient use of PROs data collected as part of regular clinical practice at a large, NCI-designated, comprehensive cancer center, as well as the utilization of statistical methods appropriate for not only examining a “yes/no” question of symptom presence but also interrogating differences in the degree and severity of PRO differences. The narrow time range for the study, and the fact that the ESAS tool was administered in specific clinical programs, also increased the likelihood that both patients with and without HIV were receiving similar treatment plans at this single institution. Although this likely helped to mitigate potential bias due to variability in treatment administration by HIV status, future efforts that include larger numbers of patients for each cancer type should specifically adjust for treatment modality. Although a short period of follow-up (March to July 2020) overlapped with the global COVID-19 pandemic, which may have interfered with clinical visits or exacerbated reported symptoms, the calendar dates of symptom assessment did not differ by cancer patient HIV status. Further limitations of this analysis should be acknowledged. The small number of cancer patients with HIV precluded an adjustment for a large number of clinical variables, although adjustment for tumor site, sex, and race, as well as stratification to patients actively receiving cancer treatment, was feasible. Stage information was missing for a large proportion of patients, and we lacked details on history of HIV treatment and therefore could not ascertain the independent contributions of HIV-associated antiretroviral therapy (e.g., duration of therapy and composition of drug regimen), versus cancer treatment toxicity, as the cause of patient-reported symptoms. Future research efforts could incorporate these data, as well as information on inflammatory biomarkers or survey-assessed sexual and lifestyle behaviors, to gain a more complete understanding of HIV-associated PRO differences during cancer care. Finally, we analyzed the worst ESAS score during the 2017–2020 study period. The patients answered an average of four to five ESAS questionnaires, regardless of HIV status. This approach provided a broad picture of their post-diagnosis treatment experience but precluded us from determining whether acute symptomatology at diagnosis was resolved after the completion of cancer therapy differently in PWH. Future studies should incorporate longitudinal PROs monitoring by HIV status.

## 5. Conclusions

This study represents one of the first descriptions of PRO differences in PWH during cancer care. Among cancer patients reporting symptoms, those with HIV reported a higher frequency and severity of symptoms during care than their HIV-uninfected counterparts, which is evidence that HIV-associated differences in important clinical endpoints following a cancer diagnosis extend beyond mortality. These data should motivate an interest in non-therapeutic PROs monitoring interventions to address poor outcomes in this underserved and immunosuppressed patient population.

## Figures and Tables

**Figure 1 cancers-14-05889-f001:**
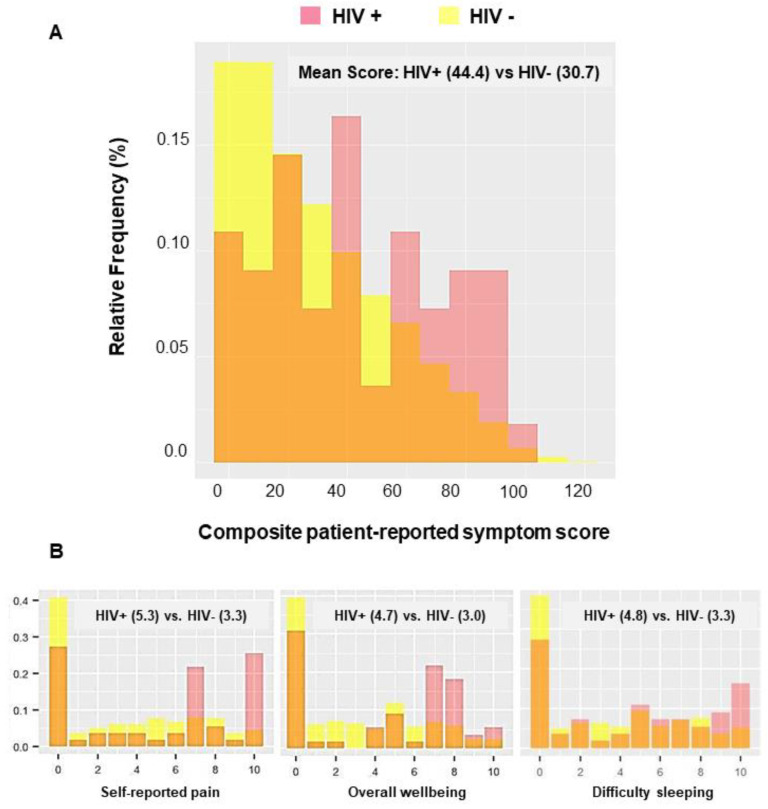
(**A**) Distribution of composite (summed) patient-reported symptom scores in cancer patients with (red) versus without (yellow) HIV. Shaded (orange) regions represent overlap in the two distributions, with red bars to the right illustrating a notable shift toward higher symptom scores in PWH and cancer. (**B**) HIV-associated patient-reported symptom differences were most pronounced for pain, overall well-being, and difficulty sleeping.

**Table 1 cancers-14-05889-t001:** Association between cancer patient HIV status and patient-reported symptoms during oncology care.

	Composite Symptom Score ^a^					Number of Severe Symptoms ^b^				
	Mean	Ratio (95% CI) for Count Model ^c^	*p*	OR (95% CI) for Zero Model ^d^	*p*	Mean	Ratio (95% CI) for Count Model	*p*	OR (95% CI) for Zero Model	*p*
Presence of HIV ^e^	44.4	1.41 (1.13–1.77)	<0.01	1.09 (0.37–3.24)	0.88	3.7	1.45 (1.09–1.93)	0.01	0.37 (0.08–1.69)	0.20
Absence of HIV	30.7	Reference		Reference		2.1	Reference		Reference	
Anal cancer	35.2	1.10 (0.99–1.21)	0.07	0.39 (0.17–0.91)	0.03	2.5	1.02 (0.88–1.19)	0.75	0.50 (0.26–0.97)	0.04
Heme malignancy	34.7	1.14 (1.07–1.22)	<0.01	0.93 (0.65–1.32)	0.67	2.4	1.16 (1.05–1.28)	<0.01	0.94 (0.71–1.25)	0.68
Other solid tumors	30.5	Reference		Reference		2.1	Reference		Reference	
Male	27.7	0.84 (0.81–0.86)	<0.01	1.99 (1.68–2.36)	<0.01	1.8	0.79 (0.76–0.83)	<0.01	1.53 (1.34–1.75)	<0.01
Female	34.4	Reference		Reference		2.5	Reference		Reference	
Black	34.1	1.11 (1.05–1.17)	<0.01	1.38 (1.06–1.8)	0.02	2.6	1.19 (1.1–1.3)	<0.01	0.97 (0.76–1.24)	0.81
Other	30.9	0.99 (0.93–1.05)	0.75	1.03 (0.75–1.41)	0.88	2.1	1.10 (1.01–1.21)	0.04	1.47 (1.17–1.83)	<0.01
White	30.5	Reference		Reference		2.2	Reference		Reference	

^a^ Each patient provided a response to 12 different symptoms (scores ranged from 0 to 10). The scores were summed (minimum 0; maximum 120). ^b^ Any symptom scoring ≥ 7 was considered as a severe symptom (minimum 0; maximum 12). ^c^ For the count model, the ratio represents the multiplicative effect of cancer patient characteristics (e.g., presence of HIV) on the mean of the respective symptom score compared to the reference group. ^d^ For the zero model, the OR represents the effect of cancer patient characteristics (e.g., presence of HIV) on the odds of reporting ‘0’ for the respective symptom score compared to the reference group. ^e^ Regression models were adjusted for cancer site (anal, heme, other solid tumors), biological sex reported in the EHR (male, female), and race reported to the cancer registry (White, Black, other).

## Data Availability

De-identified data are available from the corresponding author (A.E.C.) upon reasonable request and documentation of necessary ethical research approvals.

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
