# Peer review of "Patient-Reported Outcomes in Cancer Patients with HIV"

_cancers, 2022, doi:10.3390/cancers14235889_

Round 1

Reviewer 1 Report

This is certainly an interesting manuscript in the age of COVID-19. Patients with different immunodeficiencies, HIV among them have suffered the most. PROs has been utilized in this study appropriately with a significant difference shown between HIV infected cancer patients and non HIV infected. Though no sexual activity report is given on the generally younger male HIV infected, this could also likely contribute to inflammation associated with therapy in anal cancer patients.

The very low sample size of HIV infected and the single center study are draw back that can be addressed in future studies. 

Author Response

We thank the reviewer for their comments; please see responses in bold text.

Reviewer Comment #1: This is certainly an interesting manuscript in the age of COVID-19. Patients with different immunodeficiencies, HIV among them have suffered the most. PROs has been utilized in this study appropriately with a significant difference shown between HIV infected cancer patients and non HIV infected. Though no sexual activity report is given on the generally younger male HIV infected, this could also likely contribute to inflammation associated with therapy in anal cancer patients.

We thank the reviewer for their positive comment and agree that future studies could collect biomarker data (e.g., inflammatory markers) and more detailed survey instrumentation (e.g., sexual activity) to more completed address HIV-associated PRO differences. We have included text to this effect in the revised Discussion section starting at line 206.

Reviewer Comment #2: The very low sample size of HIV infected and the single center study are draw back that can be addressed in future studies. 

We agree and have cited this (lines 195 and 200) in our Limitations.

Reviewer 2 Report

Interesting study assessing PRO in cancer patients with and without HIV. The authors present a well organized and clearly written manuscript. Findings are important for HIV cancer care.

1. did you consider cancer stage in any of your models? Mention of sensitivity analysis restricting to early stage (line 93) but it was not adjusted for in the main findings models.

2. Were cancer treatments comparable between groups? Treatment duration similar? Lines 190-193 state that “also increased the likelihood that both patients with and without HIV were receiving similar treatment plans at this single institution, avoiding potential bias with variability in treatment administration by HIV status.” However prior publications from Dr. Suneja have shown that there are differences in cancer treatment by HIV status which led to NCCN guideline recommendations in 2018. Could there be differences in treatment which could be associated with increased morality in PWH in your study?

3. Line 137-138, can you add your comparison group to the end of that sentence

4. Interesting finding regarding “self-reported symptomology among PWH was associated with an increased risk of all-cause mortality.” Did you find similar results for patients without HIV?

Author Response

We thank the reviewer for their comments; please see responses in bold text.

Reviewer Comment #1: did you consider cancer stage in any of your models? Mention of sensitivity analysis restricting to early stage (line 93) but it was not adjusted for in the main finding's models.

Unfortunately, data were missing on stage for a large proportion of patients, particularly anal cancer patients.  As such, we reported findings for those with confirmed early-stage disease, but we have added a line to the Limitations paragraph (line 212) specifically noting this now.

Reviewer Comment #2: Were cancer treatments comparable between groups? Treatment duration similar? Lines 190-193 state that “also increased the likelihood that both patients with and without HIV were receiving similar treatment plans at this single institution, avoiding potential bias with variability in treatment administration by HIV status.” However prior publications from Dr. Suneja have shown that there are differences in cancer treatment by HIV status which led to NCCN guideline recommendations in 2018. Could there be differences in treatment which could be associated with increased morality in PWH in your study?

We have changed the language in the Discussion to emphasize the need for future research that is specific to treatment modality.  However, in addition, we have added lines 110-114 in the Results outlining a lack of difference in current administration of radiation and chemotherapy by HIV status in our patient population, with fewer surgeries in PWH. 

Reviewer Comment #3: Line 137-138, can you add your comparison group to the end of that sentence

This has been corrected.

Reviewer Comment #4: Interesting finding regarding “self-reported symptomology among PWH was associated with an increased risk of all-cause mortality.” Did you find similar results for patients without HIV?

As the focus of this report was understanding PRO differences in PWH, we focused our preliminary mortality analyses on this population that is understudied.  We do cite literature (references 24 and 25) regarding PRO monitoring as improving outcomes in the general oncology population, but that was not the focus of our report. 

Reviewer 3 Report

Major Comments

This study is well-designed and will be helpful in improving the prognosis of people living with HIV and cancer. I only have minor comments as follows:

Minor comments

Lines 58-64: Consider rephrasing this statement “Reports from the US veterans Administration (VA) indicate that anal cancer patients with HIV are more likely to report severe…”. This can be misunderstood to mean that these patients were reporting these adverse events themselves using a PRO, whereas those were reported by the study authors, which would undermine your claim as the first to undertake this study. Consider replacing “reports” with "presents" to indicate the observation of clinical events.

Add a brief description of what the relative frequency (%) refers to in Figure 1 legend.

There is a wide disparity in the sample sizes. I understand the rarity of identifying PWH Cancer individuals. I will propose subsampling the larger non-PWH Cancer+ into a comparable sample size, age, cancer type (anal cancer) as the PWH cancer individuals.

Was there any specific reason for not adjusting for age in the regression models?

Is the Odds ratio not a better statistical measure of outcomes than the ESAS score? OR for the presence of HIV though not significantly different, is 1.09 (no difference) and even less in the number of severe symptoms (Table 1). Can you explain how this fits with the narrative of this study?

In line 190-192, your hypothesis that “the narrow range for the study increases the likelihood that both patients with and without HIV were receiving similar treatment…” might be overly speculative. Even for PWH alone, the ART regimen does differ, and one can expect a more cautious and patient-specific anticancer + ART regimen in this group due to drug interactions and adverse effects. Can you confirm that both groups are on similar anticancer regimen to rule out this potential confounding factor?  

Author Response

We thank the reviewer for their comments; please see responses in bold text.

Reviewer Major Comments: This study is well-designed and will be helpful in improving the prognosis of people living with HIV and cancer. 

We thank the reviewer for this positive assessment of our work. 

Reviewer Minor comments: 

Lines 58-64: Consider rephrasing this statement “Reports from the US veterans Administration (VA) indicate that anal cancer patients with HIV are more likely to report severe…”. This can be misunderstood to mean that these patients were reporting these adverse events themselves using a PRO, whereas those were reported by the study authors, which would undermine your claim as the first to undertake this study. Consider replacing “reports” with "presents" to indicate the observation of clinical events.

This has been corrected.

Add a brief description of what the relative frequency (%) refers to in Figure 1 legend.

This will be corrected in proof-ready Figures.  The relative frequency refers to the distribution of responses by bin in the histogram.

There is a wide disparity in the sample sizes. I understand the rarity of identifying PWH Cancer individuals. I will propose subsampling the larger non-PWH Cancer+ into a comparable sample size, age, cancer type (anal cancer) as the PWH cancer individuals.

As the data are all available, we chose to use the full set of HIV-uninfected cancer patients in order to achieve the greatest amount of statistical power, while adjusting for cancer type, age, and race in adjusted regression models.  We agree that sample size is a limitation and have noted this in lines 195 and 200 in our Limitations paragraph. 

Was there any specific reason for not adjusting for age in the regression models?

Age was not related to self-reported symptomology in univariate analyses, and we prioritized having a more parsimonious model / fewer variable due to our limited sample size.  In our revised Methods, we have added information (line 86) to describe how we chose our three adjustment variables. 

Is the Odds ratio not a better statistical measure of outcomes than the ESAS score? OR for the presence of HIV though not significantly different, is 1.09 (no difference) and even less in the number of severe symptoms (Table 1). Can you explain how this fits with the narrative of this study?

The output of the "0" component of the zero inflated negative binomial (ZINB) regression models should not be interpreted as a conventional OR.  The ZINB model was chosen specifically because the Vuong closeness test, a likelihood-ratio-based method for model selection, indicated that the data best fit a scenario where the patient population was split between patients reporting no symptoms and those reporting a full range of symptoms (as opposed to none versus only 1 symptom or none versus all symptoms). If we had run a traditional OR with hypothetical categories for 'no symptoms,' 'few symptoms,' and 'many symptoms,' the cancer patients with HIV would have been preferentially distributed in the highest categories (as illustrated in our Figure 1 histograms) and therefore resulted in a positive OR.  However, that more simplistic scenario did not fit the data according to the Vuong test.  We modified the first and last paragraph of the Discussion to more accurately reflect this fact.

In line 190-192, your hypothesis that “the narrow range for the study increases the likelihood that both patients with and without HIV were receiving similar treatment…” might be overly speculative. Even for PWH alone, the ART regimen does differ, and one can expect a more cautious and patient-specific anticancer + ART regimen in this group due to drug interactions and adverse effects. Can you confirm that both groups are on similar anticancer regimen to rule out this potential confounding factor?  

We have changed the language in the Discussion to emphasize the need for future research that is specific to treatment modality.  However, in addition, we have added lines 110-114 in the Results outlining a lack of difference in current administration of radiation and chemotherapy by HIV status in our patient population, with fewer surgeries in PWH.